

# Clinical significance and heterogeneity of circulating tumor cells and clusters in breast cancer subtypes

LiangYu Hao and Lixiang Zheng

Basic Medical Research Center, Jiangxi University of Chinese Medicine, Nanchang City, Jiangxi Province, China

## ABSTRACT

The marked heterogeneity of breast cancer results in substantial variations in clinical characteristics, metastatic patterns, and prognosis across molecular subtypes. However, circulating tumor cells (CTCs) and circulating tumor cell clusters (CTC clusters), pivotal mediators of metastasis, have not been comprehensively evaluated for their biological characteristics and clinical significance across molecular subtypes. This review synthesizes recent research advancements to comprehensively examine the distribution characteristics, biological functions, and prognostic associations of CTCs and CTC clusters in luminal A, luminal B, HER2-positive breast cancer, and triple-negative breast cancer (TNBC). It was observed that HER2-positive breast cancer is associated with elevated CTC counts, whereas TNBC, despite lower CTC counts, exhibits CTCs and CTC clusters with enhanced invasiveness and metastatic potential due to Notch1 signaling pathway activation, elevated PD-L1 expression, and desialylation modifications. In luminal subtypes, the scarcity of CTC clusters is linked to a reduced metastatic risk; however, luminal B exhibits a greater propensity for CTC cluster formation than luminal A, suggesting prognostic differences. Clinical data demonstrate that CTC cluster counts are significantly inversely correlated with overall survival (OS) and disease-free survival (DFS), and that dynamic monitoring of CTC clusters enables prediction of treatment resistance and recurrence risk. Furthermore, the molecular profiles of CTCs (*e.g.*, HER2 status, ESR1 mutations) facilitate personalized guidance for targeted and endocrine therapies. However, current detection technologies exhibit limitations in capturing CTC clusters with high efficiency and sensitivity, necessitating further optimization through microfluidic sorting, single-cell omics, and artificial intelligence approaches. This review underscores the heterogeneity of CTCs and CTC clusters across breast cancer subtypes, alongside their potential for clinical translation, offering theoretical support for prognostic evaluation and individualized treatment strategies in precision medicine. This study may be of considerable value to researchers and clinicians in the field of cancer metastasis.

## INTRODUCTION

Breast cancer, ranked as the second most prevalent cancer among women and the fifth leading cause of cancer-related mortality globally (*Sung et al., 2021*), is characterized by

Corresponding author
Lixiang Zheng, 2992699831@qq.com

pronounced heterogeneity. Its heterogeneity is manifested not only in the morphological and molecular characteristics of tumors (*Hsiao et al., 2010*), but also in clinical manifestations, metastatic patterns, and therapeutic responses (*Lacruz et al., 2025*). Moreover, distinct molecular subtypes exhibit marked differences in clinical characteristics and survival outcomes (*Garcia-Recio et al., 2025*).

Notwithstanding significant differences in treatment and prognosis among breast cancer subtypes, metastasis remains the primary cause of disease progression (*McGinnis et al., 2024*). The presence of CTCs and CTC clusters signifies the dissemination of tumors from the primary site to distant organs, thereby providing a crucial perspective for investigating breast cancer metastasis. CTC clusters, defined as aggregates composed of multiple CTCs or of CTCs interconnected with other cells *via* intercellular junctions, are recognized for their distinct biological properties. Compared with individual CTCs, CTC clusters demonstrate markedly enhanced metastatic potential. Studies have demonstrated that the metastatic capacity of CTC clusters is 20–50 times greater than that of individual CTCs (*Wang, Zhou & Hu, 2017*), primarily attributable to their enhanced ability to withstand shear forces in the circulatory system and to evade immune surveillance. Moreover, CTC clusters exhibit greater genetic diversity, which facilitates superior adaptation to the microenvironment and enhances metastatic efficiency (*Gu, Wei & Lv, 2024*).

Although the roles of CTCs and CTC clusters have been extensively examined, the existing literature predominantly focuses on their impact in overall breast cancer populations, thereby leaving their specific characteristics, quantitative variations, and clinical significance across distinct molecular subtypes largely unexplored. For instance, given the heightened invasiveness associated with TNBC and HER2-positive subtypes, it is pertinent to inquire whether these subtypes are more predisposed to forming CTC clusters. Similarly, whether the infrequent CTC clusters observed in luminal A breast cancer retain prognostic significance remains an open question. Addressing these questions is imperative for elucidating the biological differences among breast cancer subtypes and for optimizing personalized treatment strategies.

The purpose of this review is to: (1) consolidate current research on CTCs and CTC clusters in breast cancer, with a focus on their distribution patterns and prognostic significance across distinct molecular subtypes; (2) analyze the association between CTCs/CTC clusters and both DFS and OS, as well as their interactions with other prognostic markers; and (3) elucidate the potential of CTCs and CTC clusters as biomarkers for the personalized diagnosis and treatment of breast cancer, thereby providing a foundation for subsequent clinical research and therapeutic interventions.

## SURVEY METHODOLOGY

Primary and secondary literature relevant to the topic of this review was assessed using PubMed (MeSH), MedLine, Google Scholar, and Web of Science using the search terms for articles and their combinations in English to search: "circulating tumor cells", "CTC clusters", "breast cancer subtypes",, "prognostic significance", "clinical research", "triple-negative breast cancer", "HER2-positive breast cancer", "Luminal A", "Luminal B",

**Table 1  Comparison of characteristics of molecular subtypes of breast cancer.**

| Molecular subtype | ER | PR | HER2 | Ki67 | Prognosis | Treatment strategies |
|---|---|---|---|---|---|---|
| Luminal A | (+) | (+) | (−) | Low (<20%) | Good. Slow-growing, lower grade, and less likely to recur | Hormone therapy |
| Luminal B | (+) | (+)/(−) | (+)/(−) | High (>20%) | Moderate. Faster-growing than Luminal A, with a slightly worse prognosis | Hormone therapy + chemotherapy, sometimes combined with anti-HER2 therapy |
| HER2+ | (−) | (−) | (+) | Variable | Poor. Aggressive and fast-growing, but treatable with targeted therapy | Anti-HER2 targeted therapy, chemotherapy |
| TNBC | (−) | (−) | (−) | Variable but often high | Worst, Aggressive and highly proliferative, with limited treatment options | Chemotherapy based, combined with targeted therapy and immunotherapy |

"breast cancer", "clinical significance", "personalized medicine", "immunotherapy", "chemotherapy", "drug resistance" and "metastasis", using operators "+", "AND", and "OR" to refine results. A preliminary examination was conducted to determine whether the article met the theme of this review. Further inspection was carried out to determine the credibility of the article content and ensure that no bias would be generated. Earlier literature reviews on the same topic were consulted to ensure key topics were not missed.

## Different subtypes of breast cancer

The currently accepted clinical classification of breast cancer subtypes is based on molecular-level differences, categorizing them into four major subtypes: luminal A, luminal B, HER2-positive breast cancer, and TNBC. This classification is primarily based on the expression levels of hormone receptors (ER, PR), human epidermal growth factor receptor 2 (HER2), and the cell proliferation marker Ki-67 (*Orrantia-Borunda et al., 2022*). The comparison of the four molecular subtypes of breast cancer is shown in Table 1.

Clinical observations and experimental studies have demonstrated marked differences in both invasiveness and clinical prognoses among various breast cancer subtypes (*Prat et al., 2015*). Hematogenous metastasis constitutes a principal pathway for the dissemination of breast cancer, with CTCs and CTC clusters playing a pivotal role in this process (*Sayed et al., 2024*). Therefore, it is reasonable to hypothesize that CTCs and CTC clusters display distinct quantitative and biological characteristics across various breast cancer subtypes, potentially contributing to enhanced diagnostic accuracy and the development of personalized treatment strategies. This review comprehensively explores this aspect, thereby offering novel insights for future research and personalized clinical management.
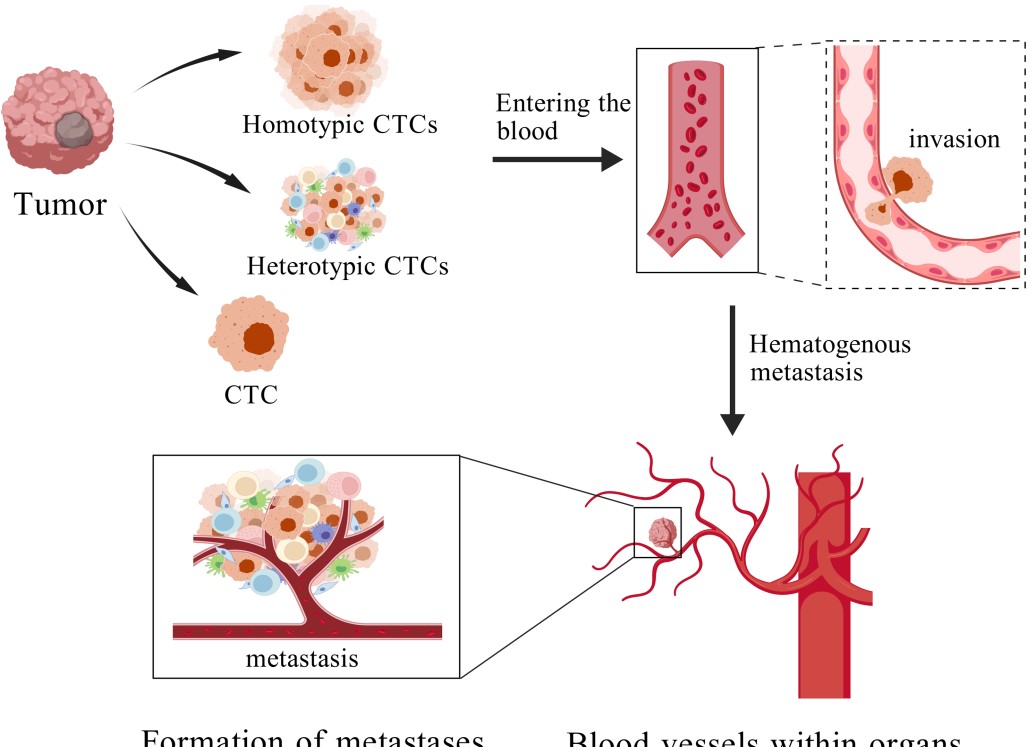

Tumor

Homotypic CTCs

Heterotypic CTCs

CTC

Entering the blood

invasion

Hematogenous metastasis

metastasis

Formation of metastases

Blood vessels within organs

**Figure 1** **The process of cancer metastasis.** Created with BioGDP.com.

## Research status of CTCs and CTC clusters
### Biological functions and heterogeneity of CTCS

Metastasis represents the most fatal manifestation of breast cancer and constitutes a primary determinant of patient mortality and poor prognosis (*McGinnis et al., 2024*). Cancer metastasis is a complex process encompassing the detachment of cancer cells from the primary site, their intravasation into the circulatory system, extravasation from the bloodstream, and subsequent colonization at distant metastatic sites (*Liu et al., 2024*). CTCs denote cancer cells that detach from the primary tumor and extravasate into the bloodstream (Fig. 1) (*Jiang et al., 2025*). CTCs are regarded as pivotal mediators of cancer metastasis (*Eslami et al., 2022*). Although originating from the primary tumor, these cells may undergo alterations in gene expression, molecular characteristics, and biological behavior upon entering the bloodstream (*Jie, Zhang & Xu, 2017*), thereby enhancing their adaptability and metastatic potentia (*Zhan et al., 2023*; *Cohen et al., 2023*).

The formation of CTCs encompasses two critical stages. The first stage, detachment, occurs when tumor cells undergo epithelial–mesenchymal transition (EMT), leading to the downregulation of adhesion molecules such as E-cadherin, thereby facilitating their release from the primary tumor into the bloodstream. The second stage, survival, necessitates that CTCs evade immune clearance, withstand oxidative stress, and resist shear forces within the bloodstream, thereby ensuring their viability and eventual establishment of distant

metastases (*Li et al., 2023*). Accordingly, the survival capacity and metastatic potential of CTCs exert a direct influence on cancer progression and patient prognosis.

Recent studies show that CTCs are not a uniform cell type but a highly diverse population, varying significantly in shape, size, gene expression, and behavior (*Pereira-Veiga et al., 2022*). This heterogeneity not only affects the survival rates, metastatic potential, and drug resistance of CTCs, but is also closely associated with patient prognosis and the formulation of personalized treatment strategies. The heterogeneity of CTCs is primarily manifest in their morphological and molecular diversity (*Gu, Wei & Lv, 2024*), and such heterogeneity directly influences the metastatic process, patient prognosis, and the selection of therapeutic strategies.

### Biological functions of CTC clusters and comparison with individual CTCs

CTC clusters typically arise through the aggregation of multiple tumor cells and the collective shedding from *in situ* carcinoma (*Yang et al., 2024*), In addition to tumor cells, these clusters may also contain red blood cells, lymphocytes, leukocytes, platelets, and cancer-associated fibroblasts (*Pereira-Veiga et al., 2022*). CTC clusters are classified into homotypic and heterotypic clusters based on the presence of non-neoplastic cells (*Aceto, 2020*). Homotypic clusters are defined as multicellular aggregates composed of two or more CTCs interconnected *via* cell–cell junctions, with their prevalence increasing as the disease progresses (*Kurzeder et al., 2025*). Heterotypic clusters comprise cellular assemblies that contain one or more CTCs in conjunction with non-malignant stromal cells (*e.g.*, leukocytes, fibroblasts, endothelial cells) or immune cells, frequently incorporating platelets (*Pereira-Veiga et al., 2022*). Such heterotypic clusters further enhance immune surveillance evasion and vascular wall adhesion through synergistic interactions with blood components, thereby promoting metastatic niche formation (*Heeke et al., 2019*).

Although CTC clusters are considerably less abundant in circulation than individual CTCs, studies have demonstrated that their metastatic potential is markedly elevated—approximately 25 to 50 times greater than that of single CTCs. CTC clusters depend on intercellular adhesion molecules, including E-cadherin and N-cadherin, to maintain cohesion (*Liu et al., 2021b*). Morphologically, they exhibit greater structural complexity and demonstrate enhanced resistance to shear forces within the bloodstream (*Gu, Wei & Lv, 2024*). Furthermore, owing to intercellular cooperation, CTC clusters exhibit heightened survival capacity and enhanced immune evasion within circulation. Additionally, their three-dimensional structure and hypoxic microenvironment facilitate colonization at distant metastatic sites (*Donato et al., 2020*).

CTC clusters diverge from individual CTCs primarily in their cellular composition, prevalence, metastatic and invasive potential, survival capacity, and drug resistance. A comparative analysis of CTCs and CTC clusters is presented in Table 2.

### Clinical detection of CTCs and CTC clusters

The detection of CTCs typically comprises two primary stages: enrichment and separation, followed by identification and analysis (*Fabisiewicz & Grzybowska, 2017*). Conventional methodologies include immunomagnetic bead capture and physical separation techniques. Immunomagnetic capture employs antibodies conjugated to magnetic beads that recognize

**Table 2  Comparison of CTC and CTC clusters.**

| Types | CTC clusters | CTCs |
|---|---|---|
| **Formation** | Form through aggregation or interaction with other blood cells | Detach as single cells |
| **Metastatic efficiency** | High | Low |
| **Survival ability** | Survive better due to protective structure | Vulnerable to damage and immune cells |
| **Invasive ability** | High | Low |
| **Drug resistance** | High | Low |

surface markers such as EpCAM (*e.g.*, the CellSearch system) (*He et al., 2024*), However, due to tumor cell heterogeneity and EMT, EpCAM expression may be downregulated, thereby compromising detection sensitivity (*Hwang et al., 2024*). In contrast, physical separation methods, such as the ISET and Ficoll techniques, facilitate CTC enrichment independent of specific surface markers (*Stamatakis et al., 2021*; *Wang et al., 2024*), thus avoiding marker-related detection loss associated with EMT. Nonetheless, these approaches often require specialized equipment and may compromise cellular integrity, potentially interfering with downstream molecular analyses (*Paoletti et al., 2019*).

Beyond physical and immunoaffinity-based methods, molecular biology techniques—such as PCR, single-cell sequencing, and fluorescence *in situ* hybridization (FISH) (*Li, Wu & Bai, 2018*; *Ma et al., 2024*; *Smilkou et al., 2024*)—are utilized to investigate gene mutations (*e.g.*, HER2, PIK3CA) and gene expression profiles in CTCs (*Gasch et al., 2016*), These technologies enable high-sensitivity molecular typing and dynamic monitoring of drug resistance, thereby informing targeted therapeutic decisions (Table 3).

Current detection platforms for CTC clusters largely mirror those used for single CTCs. However, owing to their distinct biological characteristics, CTC clusters offer greater utility in clinical prediction and monitoring of cancer progression. For instance, in patients with stage IV breast cancer, the presence of more than five CTC clusters per 7.5 mL of blood is frequently correlated with aggressive disease phenotypes, suggesting that CTC cluster detection may serve as a prognostic indicator of disease progression (*Zhang et al., 2024*).

Nonetheless, the detection of CTC clusters is impeded by several challenges. One such challenge is detection sensitivity—owing to high heterogeneity and loss of EpCAM expression during EMT, traditional immuno-enrichment techniques often fail to recover CTC clusters effectively (*Hwang et al., 2024*). Another limitation lies in current enrichment techniques: due to their larger size and increased rigidity, CTC clusters are poorly captured by single-cell enrichment methods such as CellSearch. Optimization strategies have included physical enrichment methods such as microfluidic sorting and filtration-based separation, which selectively isolate CTC clusters based on size and mechanical properties (*Mishra et al., 2025*). Emerging approaches incorporating machine learning, such as Smart-Seq2 integration, have further improved detection specificity (*Pastuszak et al., 2024*).

Despite recent advances, the effective separation and detection of CTC clusters remain challenging. Future investigations may focus on developing multi-dimensional detection models that integrate CTC clusters with other biomarkers (*e.g.*, cfDNA, exosomes) to

**Table 3 Separation and identification methods of CTC and CTC clusters.**

| Method | Principle | Advantages | Limitations | Representative products/ applications | References |
|---|---|---|---|---|---|
| **Filtration** | Size-based sieving: CTCs, being larger than most blood cells, are retained on a microporous membrane | 1. Label-free, high throughput 2. Simple and cost-effective 3. Preserves cell viability and morphology, enabling capture of multicellular clusters | 1. May miss small or highly deformable CTCs 2. Membrane fouling and clogging 3. Co-enrichment of large leukocytes reduces purity | ISET | *Vona et al. (2000)*, *Leitão et al. (2023)* |
| **Microfluidic size/inertia** | Microchannel or spiral-based inertial focusing separates cells according to size and hydrodynamic properties | 1. Label-free; maintains cell viability 2. Continuous, automated processing of large volumes 3. Gentle on CTC clusters, preserving cluster integrity | 1. Requires sample dilution or preprocessing 2. Channel clogging risk 3. High device complexity and cost | Parsortix | *Karabacak et al. (2014)*, *Low & Wan Abas (2015)*, *Ozkumur et al. (2013)* |
| **Acoustophoresis/ Dielectrophoresis** | Acoustic or nonuniform electric fields exploit differences in compressibility or polarizability | 1. Label-free; highly gentle separation 2. Can achieve very high capture efficiencies (>94%) 3. Fine-tuned selectivity | 1. Low throughput 2. Complex, expensive instrumentation 3. Limited clinical translation | DEPArray Menarini), | *Gossett et al. (2010)* |
| **Density-gradient centrifugation** | Centrifugation in Ficoll or Onco-Quick gradients enriches CTCs and mononuclear cells at the same interface | 1. Simple and inexpensive 2. Can process large blood volumes; typical recovery >80% | 1. Low purity due to co-isolation of monocytes 2. Centrifugal forces may impair cell function | Ficoll-Paque™ Plus, OncoQuick tubes | *Low & Wan Abas (2015)*, *Hou et al. (2013)*, *Diamond et al. (2012)* |
| **Positive immunoaffinity** | Antibody-mediated capture of CTCs *via* surface antigens (*e.g.*, EpCAM, HER2) | 1. High specificity; FDA/CE-approved systems available 2. Direct integration with immunocytochemistry or fluorescence assays | 1. Misses CTCs undergoing EMT with low marker expression 2. Dependence on antibody quality 3. Higher reagent cost | CellSearch, Microfluidic CTC-Chip | *Riethdorf et al. (2007)*, *Pantel & Alix-Panabières (2019)* |
| **Negative immunoaffinity** | Removal of CD45+ leukocytes *via* magnetic beads enriches all remaining non-leukocyte cells, including CTCs | 1. Marker-independent capture of all CTC phenotypes 2. Retains rare or atypical CTCs | 1. Residual leukocyte contamination lowers purity 2. Multistep protocols increase complexity 3. Potential nonspecific loss of CTCs | CTC-iChip (NIH), Dynabeads CD45 Depletion | *Warkiani et al. (2014)* |

enhance the diagnostic accuracy of breast cancer (*Khandare et al., 2024*). Additionally, the incorporation of artificial intelligence technologies to optimize automated detection platforms, alongside clinical validation, represents a critical avenue for future investigation.

## Association of CTCs and CTC clusters with breast cancer subtypes
### *The frequency and quantity of CTCs and CTC clusters vary among breast cancer subtypes*

Current research suggests that the number of CTCs and CTC clusters varies substantially among breast cancer subtypes and is strongly correlated with their respective invasiveness and metastatic potential. A clinical trial comparing HER2-positive and luminal A patients has demonstrated elevated CTC counts in the HER2-positive cohort, potentially attributable to HER2 gene amplification and the activation of signaling pathways that facilitate EMT. HER2 overexpression may enhance tumor cell proliferation, invasiveness, and endothelial translocation, consequently resulting in an increased release of CTCs into the circulation (*Zhang et al., 2021*).

In luminal A-type breast cancer, CTC and CTC cluster counts are generally limited, which may underlie its reduced invasiveness and metastatic capacity. In contrast, luminal B-type breast cancer exhibits markedly increased levels of CTCs and CTC clusters (*Smerage*

*et al., 2014*). supporting the hypothesis that luminal A tumors demonstrate lower metastatic aggressiveness compared to luminal B subtypes (*Chai et al., 2023*).

Notably, although TNBC is recognized for its pronounced invasiveness and metastatic behavior, several studies have paradoxically reported lower CTC counts in this subtype (*Munzone et al., 2012*), a trend corroborated by additional clinical reports (*Costa et al., 2020*). This paradox may be explained by the higher invasiveness of TNBC-derived CTCs or their increased propensity to form clusters. Given that CTC clusters exhibit significantly greater metastatic potential than individual CTCs, this may account for the elevated metastatic propensity of TNBC despite lower overall CTC counts.

To date, no large-scale, systematic clinical studies have quantitatively compared CTC and CTC cluster levels across the four major molecular breast cancer subtypes. Future research should incorporate multi-center clinical trials and leverage advanced techniques—such as single-cell sequencing and circulating biomarker profiling—to elucidate the quantitative characteristics and clinical relevance of CTCs and CTC clusters in different breast cancer subtypes. Furthermore, investigating whether TNBC exhibits a greater tendency to form CTC clusters, and the extent to which this contributes to its metastatic aggressiveness, represents an important avenue for future investigation.

### Biological functions of CTCs and CTC clusters in different subtypes

Significant differences in metastatic potential exist across various breast cancer subtypes, and the biological functions of CTCs and CTC clusters may play a crucial role in these variations. In the highly metastatic TNBC subtype, CTCs and CTC clusters exhibit elevated Notch1 signaling pathway activity, which substantially promotes EMT, thereby enhancing their motility and invasiveness and facilitating distant metastasis (*Boral et al., 2017*). Additionally, CTCs in TNBC patients display elevated PD-L1 expression, which interacts with the PD-1 receptor on T cells, effectively suppressing T cell–mediated anti-tumor immunity and permitting immune evasion (*Vardas et al., 2023*). Moreover, studies indicate that CAIX—a target gene of hypoxia-inducible factor 1α (HIF1α)—is markedly upregulated in TNBC and HER2-positive breast cancer cell lines, playing a critical role in promoting CTC survival, immune evasion, and drug resistance(*Twomey & Zhang, 2023*). Regarding CTC cluster formation, desialylation of CTCs in TNBC promotes intercellular adhesion by exposing galactose residues—for example, *via* galectin-3–mediated aggregation—thereby enhancing cluster formation and chemotherapy resistance(*Glinsky et al., 2003*; *Gvozdenovic & Aceto, 2023*). Simultaneously, elevated expression of epidermal growth factor receptor (EGFR) effectively promotes CTC cluster formation and enhances their stability (*Liu et al., 2021a*). Notably, CTCs detected in TNBC patients often exhibit a TLR4+/pSTAT3+ phenotype, with Toll-like receptor 4 (TLR4) and phosphorylated signal transducer and activator of transcription 3 (pSTAT3) playing pivotal roles in tumor cell proliferation, invasion, and migration (*Papadaki et al., 2022*). Collectively, these factors contribute to the increased invasiveness and drug resistance of CTCs and CTC clusters in TNBC, thereby accelerating distant metastasis.

HER2-positive breast cancer represents one of the most metastatic subtypes, with CTCs displaying higher Ki67 expression—indicative of enhanced proliferative activity

and survival capacity (*Boral et al., 2017*). Regarding cell migration, HER2-positive CTCs may demonstrate increased migratory and invasive capabilities during the EMT process. Moreover, activation of the HER2 signaling pathway may promote EMT and further enhance CTC survival and migration by stimulating downstream pathways such as PI3K/AKT and MAPK (*Bulfoni et al., 2016*). Additionally, overexpression of the HER2 receptor is common in both HER2-positive CTCs and CTC clusters, which not only supports cell proliferation and survival but may also enhance intercellular adhesion, thereby improving the ability of CTC clusters to resist shear forces and evade immune surveillance in circulation (*Nasr & Lynch, 2023*).

Luminal-type breast cancer is classified into luminal A and luminal B subtypes based on hormone receptor expression and cellular characteristics. Studies have demonstrated that CTCs in luminal-type breast cancer predominantly exhibit an epithelial phenotype, with both CTCs and CTC clusters displaying relatively lower invasiveness (*Bulfoni et al., 2016*). In luminal A breast cancer, CTCs demonstrate lower PD-L1 expression (*Papadaki et al., 2021*). Combined with relatively weak estrogen receptor (ER) signaling, this may result in a shortened survival time of CTCs in the bloodstream, thereby reducing their metastatic potential (*Koch et al., 2020*). In contrast, in luminal B breast cancer, the CTC count and the formation rate of CTC clusters are generally higher than in luminal A–type, potentially contributing to its greater metastatic potential (*Holler et al., 2023*).

Overall, these studies reveal significant differences in the invasiveness, survival mechanisms, and immune evasion capabilities of CTCs and CTC clusters across various breast cancer subtypes, providing novel insights into their metastatic characteristics. However, research on the biomarkers and immune evasion mechanisms of CTCs and CTC clusters across the four subtypes remains limited. Future studies should further investigate the mechanisms underlying CTC cluster formation and their roles in metastasis and drug resistance, as well as evaluate the clinical utility of CTC-related biomarkers in precision medicine for breast cancer, thereby enhancing our understanding of the relationships among different breast cancer subtypes.

## Clinical significance of CTCs and CTC clusters in breast cancer patients

### Relationship between CTCs and CTC clusters and prognosis of breast cancer patients

It is currently recognized that the quantity of CTCs and CTC clusters is closely associated with OS and DFS in patients with breast cancer. Numerous studies have demonstrated that elevated CTC counts are correlated with poorer prognostic outcomes, and the presence of CTC clusters further exacerbates the risk of disease progression and metastasis. Notably, CTC clusters possess greater predictive value for recurrence risk compared to individual CTCs in breast cancer patients. Existing evidence indicates that the metastatic efficiency of CTC clusters is approximately 50 times higher than that of solitary CTCs (*Schuster et al., 2021*), Moreover, reductions in CTC counts observed during treatment have been associated with enhanced therapeutic responses and improved survival outcomes, whereas persistent or increasing numbers of CTC clusters may reflect therapeutic resistance and

**Table 4** Comparison of CTC and CTC cluster counts across breast cancer subtypes.

| Breast cancer subtype | CTC count | CTC cluster count | Association with poor prognosis |
|---|---|---|---|
| Luminal A | Detection rate: ∼60%; Median: 2 cells/7.5 mL blood | Low frequency; Median: 4 clusters | Yes (CTC positivity correlates with recurrence and shortened survival) |
| Luminal B | High detection rate: ∼90%; Median: 2 cells/7.5 mL blood | Low frequency; Median: 4 clusters | Yes (CTC positivity correlates with recurrence and shortened survival) |
| HER2 + | Very high detection rate: ∼97%; Median: 4 cells/7.5 mL blood | Rare; Median: 0 clusters (detected in ∼45% of patients) | Yes (CTC positivity often indicates disease progression) |
| TNBC | Extremely high detection rate: ∼100%; Median: 2.5 cells/7.5 mL blood | Frequent; Median: 5 clusters | Yes (CTC positivity significantly correlates with reduced OS and PFS) |

indicate a heightened likelihood of tumor recurrence (*Costa et al., 2020*), Collectively, these findings underscore the critical role of CTC clusters in the clinical assessment of OS and DFS, as well as in prognostic evaluation.

### Differences in the number of CTCs and CTC clusters between different subtypes of breast cancer

In clinical investigations, significant heterogeneity has been observed in both the detection rate and quantity of CTCs and CTC clusters across the molecular subtypes of breast cancer. Specifically, patients with luminal A tumors exhibit a CTC detection rate of approximately 60%, with the majority demonstrating low CTC counts (median: two cells per 7.5 mL of blood) (*Zhou et al., 2022*). CTC clusters are exceptionally rare in this subtype (median cluster count: 4) (*Sayed et al., 2024*). In contrast, patients with luminal B subtype demonstrate a CTC detection rate approaching 90% (*Xu et al., 2018*), though with a similar CTC count (median: 2 cells/7.5 mL) and cluster frequency to luminal A (*Reduzzi et al., 2021*). Notably, HER2-positive patients exhibit near-universal CTC detection (≈97%) with a median CTC count of 4 cells/7.5 mL (*Xu et al., 2018*), while CTC clusters are detected in 45% of cases (median: 0; range: 0–8) (*Reduzzi et al., 2021*). Of particular interest, TNBC patients universally present detectable CTCs (median: 2.5 cells/7.5 mL) along with the highest median cluster count (approximately 5) (*Xu et al., 2018*) (Table 4). These findings collectively suggest that CTC clusters occur more frequently in luminal subtypes and TNBC compared to HER2-positive disease, whereas HER2-positive tumors are associated with both higher CTC counts and detection rates than luminal subtypes. Critically, CTC enumeration (including cluster quantification) across all four subtypes demonstrates significant correlation with adverse clinical outcomes (*Aceto et al., 2014*; *Gkountela et al., 2019*).

### The value of CTCs and CTC clusters in the treatment of different subtypes of breast cancer

Studies have demonstrated that the detection of HER2-positive CTCs is associated with reduced DFS in patients, suggesting that CTCs may serve as early indicators of resistance to anti-HER2 therapies (*Masuda et al., 2016*). In HER2-positive breast cancer, the continued expression of HER2 in CTCs may indicate a favorable response to trastuzumab treatment; in

contrast, the presence of HER2-negative CTCs may reflect increased therapeutic resistance (*Zhang et al., 2024*).

In patients with TNBC, the detection of PD-L1-positive CTCs has been positively correlated with the therapeutic efficacy of PD-1/PD-L1 immune checkpoint inhibitors, such as atezolizumab, indicating that PD-L1-positive CTCs may function as predictive biomarkers for identifying likely responders (*Vardas et al., 2023*). Similarly, in luminal-type breast cancer, mutations in the estrogen receptor 1 (ESR1) gene detected in CTCs have been strongly associated with endocrine resistance. These mutations induce conformational changes in the estrogen receptor (ER), allowing for ligand-independent activation of ER signaling, thereby diminishing the effectiveness of anti-estrogen therapies such as tamoxifen or letrozole (*Fridrichova, Kalinkova & Ciernikova, 2022*). As such, ESR1-mutant CTCs may serve as predictive markers of endocrine therapy resistance and could inform early treatment adaptation to agents such as fulvestrant or cyclin-dependent kinase 4/6 (CDK4/6) inhibitors (*Brett et al., 2021*).

Additionally, gene expression analyses (GEA) of CTCs have revealed dynamic changes in ER status during treatment in some patients with luminal-type breast cancer. Specifically, a phenotypic shift from ER-positive to ER-negative CTCs has been observed, which may signify emerging resistance to endocrine therapy and necessitate timely modifications to the therapeutic regimen (*Jakabova et al., 2017*).

In conclusion, CTCs and CTC clusters expressing distinct biomarkers across breast cancer subtypes possess significant potential for prognostic evaluation and personalized treatment decision-making. Further large-scale prospective studies are warranted to validate the clinical utility of CTCs and CTC clusters in the context of precision medicine for breast cancer.

## Existing problems and future prospects

Future research on CTCs and CTC clusters across different molecular subtypes of breast cancer holds considerable promise. Despite notable advances in the detection and characterization of CTCs, significant challenges remain in their clinical implementation. Firstly, although current detection technologies—such as cell sorting, liquid biopsy, and single-cell sequencing—have markedly improved in terms of sensitivity and specificity, key obstacles persist in enhancing the enrichment efficiency of CTC clusters and in establishing precise detection strategies tailored to specific molecular subtypes of breast cancer. Presently, widely adopted clinical enrichment systems, such as the EpCAM-dependent CellSearch platform, demonstrate limited capture efficiency for CTCs undergoing EMT, potentially leading to the omission of more aggressive CTC subpopulations, particularly in triple-negative breast cancer. Recently developed label-free inertial microfluidic techniques have demonstrated broad-spectrum capture capabilities for CTCs with heterogeneous surface marker expression; however, their clinical utility remains to be comprehensively validated (*Deng et al., 2017*; *Zhu et al., 2024*). Future research should prioritize the refinement of CTC and CTC cluster isolation methodologies—such as microfluidic chip-based platforms and high-throughput single-cell analyses—to enhance the accuracy and reliability of clinical diagnostics.

Secondly, further investigation is required to elucidate the mechanisms underlying the formation and functional roles of CTCs and CTC clusters in metastasis across various breast cancer subtypes. The presence of CTCs and CTC clusters is closely associated with patient prognosis, and their biological features—including mesenchymal phenotypes, genetic mutations, and epigenetic alterations—may play direct roles in promoting tumor cell invasion and metastasis. However, the quantitative and qualitative heterogeneity of CTCs, as well as the composition and functional behavior of CTC clusters among different molecular subtypes, remain insufficiently characterized. Future studies should incorporate emerging technologies such as cytomics, spatial transcriptomics, and proteomics to elucidate the molecular mechanisms that govern these processes.

Additionally, the clinical feasibility and translational potential of CTCs and CTC clusters in breast cancer management require further empirical validation. Key questions remain: Can dynamic fluctuations in CTC populations serve as reliable biomarkers for predicting therapeutic responses and disease progression? Are CTCs and CTC clusters viable tools for guiding personalized treatment decisions across distinct molecular subtypes? How can artificial intelligence (AI) and machine learning algorithms be integrated to enhance the accuracy of CTC detection and data interpretation? Addressing these challenges will be essential for bridging the gap between laboratory-based discoveries and real-world clinical applications, ultimately improving treatment efficacy and patient survival outcomes.

In conclusion, future investigations into CTCs and CTC clusters in breast cancer should focus on three primary domains: technological innovation, mechanistic elucidation, and clinical translation. Through the integration of multidisciplinary approaches, including bioengineering, artificial intelligence, and precision oncology, the clinical utility of CTCs in breast cancer diagnosis and treatment is expected to be significantly expanded, thereby accelerating progress in the field of precision medicine.

## CONCLUSION

This review systematically evaluates the clinical significance of CTCs and CTC clusters across distinct molecular subtypes of breast cancer, with a focus on their critical roles in disease progression, metastasis, and monitoring of treatment response. By analyzing both the biological characteristics and quantitative disparities of CTCs and CTC clusters among breast cancer subtypes, this review highlights the inherent heterogeneity of these subtypes and offers novel insights that support the advancement of personalized medicine.

First, substantial differences in CTC and CTC cluster profiles have been reported among breast cancer subtypes. For example, HER2-positive patients exhibit higher CTC counts, while luminal A and luminal B subtypes are characterized by relatively lower levels of both CTCs and clusters. Although TNBC patients often have fewer CTCs, their CTC clusters exhibit significantly greater invasiveness and metastatic capacity, suggesting that clusters may serve as more clinically relevant biomarkers in specific subtypes.

Second, the prognostic value of CTCs and CTC clusters has been extensively validated. Elevated counts of these circulating biomarkers are consistently associated with decreased OS and DFS, reinforcing their utility in recurrence risk assessment. Moreover, molecular

profiling of CTCs—such as the detection of ESR1 mutations—enables the early prediction of endocrine therapy resistance and facilitates informed adjustments to treatment strategies.

Nevertheless, several challenges remain. Current detection technologies still lack sufficient sensitivity and specificity, particularly in capturing CTCs undergoing EMT. Future research should emphasize the refinement of CTC and CTC cluster isolation techniques using advanced methodologies, including microfluidic chip platforms, high-throughput single-cell sequencing, and artificial intelligence–driven algorithms, to improve clinical applicability and diagnostic accuracy.

In conclusion, deepening our understanding of the biology of CTCs and CTC clusters, coupled with technological innovation, will be essential to establish these biomarkers as key tools in the implementation of precision medicine for breast cancer. Realizing this potential will require sustained multidisciplinary collaboration across the fields of bioengineering, artificial intelligence, and oncology to facilitate the translation of basic research into clinical practice. Moving forward, research efforts should be concentrated on three strategic pillars: technological optimization, mechanistic elucidation, and clinical validation through large-scale, prospective studies.

### Funding
This work was supported by the Key Research and Development Program of Jiangxi Province (20232ACB206053) and the Key Research and Development Program of Jiangxi Province (20203BBGL73205). The funders had no role in study design, data collection and analysis, decision to publish, or preparation of the manuscript.

### Grant Disclosures
The following grant information was disclosed by the authors:
Key Research and Development Program of Jiangxi Province: 20232ACB206053, 20203BBGL73205.

### Competing Interests
The authors declare there are no competing interests.

### Author Contributions
- LiangYu Hao conceived and designed the experiments, performed the experiments, analyzed the data, prepared figures and/or tables, authored or reviewed drafts of the article, and approved the final draft.
- Lixiang Zheng conceived and designed the experiments, analyzed the data, authored or reviewed drafts of the article, and approved the final draft.

### Data Availability
This is a literature review.

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
