# Peer review of "Clinical significance and heterogeneity of circulating tumor cells and clusters in breast cancer subtypes"

_PeerJ, doi:10.7717/peerj.19703_

## Round 0.1 · original submission · Major Revisions

The authors are requested to carefully revise the manuscript and answer the questions raised by the reviewers.

Reviewer 1 ·

Basic reporting

This review article addresses a topic of significant clinical relevance—the correlation between circulating tumor cells (CTCs) and breast cancer prognosis. Given that breast cancer remains a highly prevalent and aggressive malignancy, and CTCs serve as critical "seeds" for metastatic dissemination, the focus of this work is both timely and valuable. The authors’ comprehensive summary of current evidence provides meaningful clinical insights. However, the manuscript would benefit from improvements in some key areas according to the detail comments. The writing of this study needs to be improved before further consideration for publication.

Experimental design

The selected search strategy demonstrates adequate sensitivity to identify pertinent studies in this field.

Validity of the findings

“Novelty” While several publications on this topic are already indexed in PubMed, the authors have extended current knowledge by specifically analyzing the prognostic relevance of circulating tumor cells across different breast cancer subtypes. This subtype-stratified approach provides novel insights that advance the field.
“Value of findings” The systematic summary of the correlation between CTCs and breast cancer prognosis provides clinically actionable insights and valuable references for therapeutic decision-making.

Additional comments

1. “Language and Flow” The writing requires refinement, particularly in the use of transitional phrases. Several instances of inappropriate or abrupt transitions were noted, which may hinder readability. For example, ‘however’ on line 30 does not seem appropriate.
2. “Schematic Diagrams” The current figure lack accuracy in conveying key concepts. Optimizing visual elements (e.g., labeling, color schemes, and annotations) would improve their value.
3. Sentence on line 125-128 need more individual reference.
4. I would like to recommend that the authors provide a systematic summary of the reported quantities of circulating tumor cells (CTCs) and CTC clusters in the collected literature, along with a comprehensive analysis of their correlation with breast cancer prognosis.
5. The manuscript would benefit from professional language polishing to meet the journal's publication standards.

Reviewer 2 ·

Basic reporting

In the review by Hao and Zheng, the authors focus on the heterogeneity that characterizes CTCs and CTC clusters in breast cancer. They describe their strategy for identifying the articles for their survey, focusing on the subtypes and patterns associated with different breast cancer subtypes, prognostic features, and finally clarifying their potential as biomarkers. The topic of the review is interesting, given that in recent years, aspects have emerged that have not yet been integrated into clinical practice, aside from enumeration. However, I have a few comments I would like them to address:

The manuscript should be read through entirely and polished. There are several editing errors throughout the manuscript, and some sentences are repeated within the same paragraph, making it verbose and redundant (for example, lines 138-140; lines 141-142). I suggest increasing the level of detail and providing more studies as examples.

In section 2.2, the authors should be more specific and define homotypic clusters and heterotypic clusters. This would make the manuscript more scientifically enriched. Additionally, I propose citing some fundamental studies in the area of CTC clusters that were not mentioned in the manuscript (10.1038/s41591-024-03486-6; https://doi.org/10.1016/j.bj.2019.11.002).

Figure 1: Improve the reference to CTC clusters by providing examples of homotypic and heterotypic clusters.

One of the main reasons CTCs struggle to enter clinical practice is the difficulty in isolating them. To date, there are several approaches, each with pros and cons, making it difficult to find the right balance. The authors should emphasize the section related to CTC and CTC cluster detection, creating a very specific and easily consultable table defining the approach, pros, and cons.



Line 531: There seems to be an incorrect phrase in the references. Please check.

Experimental design

No comment

Validity of the findings

No comment

Additional comments

No comment

---

## Round 0.2 · Minor Revisions

The authors are requested to carefully revise the manuscript and answer the questions raised by the reviewers.

Reviewer 1 ·

Basic reporting

The revised manuscript meet the criteria for publication. However, I can still find some errors in the manuscript that may lead to dyslexia, such as the sentences in lines 143-145. Authors should double check that whether the descriptions in the manuscript are consistent with the original intent. As mentioned previously, the schematic diagram (Fig.1) should be more accurately expressed. Although lung metastasis is the most common site of breast cancer metastasis, the schematic of lung cannot be used to represent all metastatic organs. All requests mentioned should be addressed prior to acceptance.

Experimental design

No comment.

Validity of the findings

No comment.

Additional comments

No.

Reviewer 2 ·

Basic reporting

The manuscript can be accepted in current form.

Experimental design

NA

Validity of the findings

NA

Additional comments

NA

---

## Round 0.3 · accepted · Accept

After revisions, all reviewers agreed to publish the manuscript. I also reviewed the manuscript and found no obvious risks to publication. Therefore, I also approved the publication of this manuscript.

Reviewer 1 ·

Basic reporting

No comment

Experimental design

No comment

Validity of the findings

No comment

Additional comments

No comment